# The Effects of Neuromuscular Training on Electromyography, Lower Extremity Kinematics, and Ground Reaction Force During an Unanticipated Side-Cut on Recreational Female Hockey Players

**DOI:** 10.3390/bioengineering12101101

**Published:** 2025-10-13

**Authors:** Tom Johnston, Stephanie Valentin, Susan J. Brown, Konstantinos Kaliarntas

**Affiliations:** 1School of Applied Sciences, Sighthill Campus, Edinburgh Napier University, Edinburgh EH11 4BN, UK; su.brown@napier.ac.uk (S.J.B.); kaliarntas@upatras.gr (K.K.); 2School of Health and Social Care, Sighthill Campus, Edinburgh Napier University, Edinburgh EH11 4BN, UK; s.valentin@napier.ac.uk; 3Department of Physiotherapy, University of Patras, GR26504 Patras, Greece

**Keywords:** biomechanics, neuromuscular training, field hockey, injury prevention, side-cut

## Abstract

During an unpredictable side-cut, this study examined how a sport-specific neuromuscular training program (NMTP) influenced electromyography responses in the lower limb posterior muscles, leg movement angles, maximum vertical ground reaction force (vGRF), and the rate of force development of vGRF. Thirty-eight adult female recreational hockey players were randomly allocated into an intervention group (INT) or a control group (CON). Before beginning training or matches, the INT carried out the NMTP three times per week for eight weeks, whereas the CON performed their routine warm-up. A 45° sidecut (dominant leg only) was performed at baseline and after eight-weeks and recorded with a motion capture system. The effect of group and time, and their interaction, was investigated using a mixed-design ANOVA. After landing, the participants in the INT had greater activation of their gastrocnemius lateralis, gastrocnemius medialis, and gluteus maximus muscles than those in the CON. INT participants showed significantly lower amounts of maximum knee abduction and knee excursion, while there was an increase in these variables for the CON. At week eight, the vGRF RFD decreased for the INT but increased for the CON. Although non-significant, the overall muscle activity showed an increasing trend for the INT when it came to supervised NMTP for eight weeks compared to the effect seen in the CON. This activity caused greater alterations in the motion and forces of the lower body for the INT than the CON.

## 1. Introduction

Field hockey is a high-intensity, intermittent, invasive Olympic team sport [1]; therefore, there are a considerable number of injuries. The injury rates range from 0.1 to 90.9 per 1000 h played [2] and most injuries—up to 64%—that occur are ‘non-contact’ [2,3,4].

Most injuries in hockey occur to the lower extremity, most commonly to hamstring (0.8/1000 Athlete-Exposures) [5] and knees (0.57/Athlete-Exposures) and with ankle (0.76/1000 Athlete-Exposures) [6] and hip joints also frequently affected [2,4,7]. The injuries cause muscle strains and ligament damage [8,9]. The mechanisms involved in the most non-contact injuries are side-cutting, acceleration, deceleration and landing [9], similar to other team sports [10]. The proportion of non-contact and frequency of injuries is of concern, and therefore injury reduction mechanisms and programs require further attention [2].

Injury prevention programs have been shown to be useful in reducing injury rates in team sports [11,12,13,14,15] including hockey [16,17]. Injury prevention programs in hockey have reduced injuries (hazard ratio of 0.64) and the injury burden, a reduction of 8.42 days lost per 1000 player-hours [16] in youth field hockey players. Also, in elite female hockey players, reduced Anterior Cruciate Ligament (ACL) injury rates (0.4 per 1000 player hours in the control year) to zero in the intervention period was effective in reducing knee valgus moments and muscle activation [17].

Both the above studies used a combination of targeted training activities, known as neuromuscular training (NMT), which can improve certain characteristics (i.e., motor control, strength, etc) and potentially reduce the risk for various injuries [18]. Through its implementation, there has been a 26% decrease in all injuries, a 32% decrease in ankle sprains, a 61% decrease in ACL injuries, and reductions of up to 70% in hamstring injuries [19].

The most promising NMT programs appear to be multi-modal. Those programs, including plyometric training, improving body control (core stability), strength, balance, and sports specific movements, benefit athletes the most. The multi-modal programs that also have instructor feedback appear to produce even more effective results [20,21]. The optimal dose appears to be at least 20 min, twice a week, in order a NMT to elicit positive results [22], i.e., increasing muscle activation, which can also alter kinematics and landing forces.

Increasing muscle activation during landing through NMT can potentially reduce the injury risk [17], and Parsons [18] observed that gluteal muscle activation following NMT helped to reduce ACL injury risks by lowering hip adduction and reducing ground reaction forces when landing. A softer landing helps to reduce forces at the hip, rather than letting ligaments and similar structures do this work [23]. In field hockey, NMT programs have made positive impacts. Throughout a season, youth players who underwent an NMT intervention were less likely to suffer injuries [16]. In contrast, NMT with elite women resulted in reduced likelihood of both lower limb injuries and considerably lower knee valgus at its peak [17]. This was probably because NMT caused greater muscle activation in large muscle groups, such as the gluteal muscles, resulting in greater stability of the lower limb joints during activities such as landing.

Although there are many studies assessing the effects of NMT on females and two studies assessing the effects of NMT on hockey players, there are, to the authors knowledge, no studies assessing how NMT affects the biomechanics of recreational female hockey players during an unanticipated cut. Therefore, the aim of the current study is to investigate how an eight-week NMT program affect EMG, kinematics, and kinetics of recreational female hockey players during an unexpected side-cut. We hypothesized that implementing a NMT would result in greater muscle activity during take-off and landing (primary result), improved injury-related movement patterns (secondary result), and less force (tertiary result) during landing compared to the control group performing their usual exercises.

## 2. Materials and Methods

### 2.1. Study Design and Participants

A controlled-trial design used in this study. Experienced female university athletes who play hockey were recruited (via posters, hockey websites, and word of mouth), and those on the same team were divided into the control and intervention groups. Hockey experience, participating in training and games at least 3 times per week for at least a year, not having any injuries for 3 months, and agreeing to be part of the study were all included as eligibility requirements. If people had any current or past musculoskeletal injuries that impacted their movements, they were excluded from the study.

This study was approved under the ethical standards of Edinburgh Napier University and the School of Applied Sciences. Donnelly et al. [24] demonstrated that for the current dataset (using G*Power^®^, Los Angeles, CA, USA version 3.1.9.3), 14 subjects per group were required to provide 80% power at a 5% alpha level.

### 2.2. Intervention and Implementation

The intervention involved an evidence-based NMT program 3 times weekly for 8 weeks, presented following the TIDieR guidelines [25] (Appendix A
https://www.mdpi.com/article/10.3390/bioengineering12101101/s1). It was informed by previously published works [14,22,26,27,28,29,30,31]. The NMTP taught the players new ways to run and land, engage muscles for better mobility, strengthen their core and lower body, improve balance, and work on agility and sport specific movements. All the elements appeared in situations that were both expected and unexpected. The only equipment used was everyday hockey items: hockey sticks, balls, and cones.

The primary researcher (TJ) carried out the intervention to ensure consistency. The players received feedback, which was given using external coaching methods, to help improve their skills [32]. At the beginning of each training session and match, athletes in INT completed the Neuromuscular Training Program 3 days each week instead of performing their typical warm-up [33]. For each training session, the players were advised to complete all the exercises so that, step by step, their quality would increase and each activity would be performed vigorously (that is, rated 8 out of 10 according to Borg’s scale [34]). This was possible when the participants were guided, motivated, and given feedback by a trained instructor during the intervention [21]. During the program, the number of completed sessions was regularly recorded.

The intervention consisted of a (1) pulse-raising section with (2) muscle activation into a (3) mobilization section. The program then has a (4) core stability phase, then a (5) balance phase, followed by a (6) plyometrics and strength phase, then on to (7) agility and (8) potentiation phases. The warm-up concluded with a sport-specific movement section. The total time for the program was 20 min (Appendix A).

For the warm-up during the intervention, the control group carried on with 2 min of pulse raising, different static and dynamic stretches, and hockey-related skills and moves related to their teams’ usual activities, unlike the intervention group’s NMTP.

### 2.3. Test Protocol

All participants performed an unanticipated cut (USC) at the baseline and 8 weeks later. Every testing session began with screening (PARQ) and height and body mass measurements, followed by performing a standard warm-up and practicing the sidecut. Before week 8, every player completed a questionnaire on the amount of hockey played, physical training and record any injuries. Testing for the study took place in the biomechanics laboratory at the Sighthill Campus of Edinburgh Napier University.

### 2.4. Unanticipated Side-Cut (USC)

The USC is a frequent sports activity, and its reliability has been tested before since it is also a frequent cause of injuries [35]. The direction of travel was determined by the timing gates positioned before the force plate and all participants performed five fast 45° side-cuts towards the indicated side (indicated by SmartSpeed Pro™, Fusion Sport^®^, Highlands Ranch, CO, USA) at the highest possible speed (minimal speed included was 2.14 m/s) (Figure 1a,b).

## 3. Data Collection and Processing

### 3.1. EMG

For this study, two Delsys Trigno Wireless™ sensors (SP-W01D, Natick, MA, USA), set to 1925.925 Hz (up-sampled to 2000 Hz), were applied to Gluteus Maximus (GMax), Gluteus Medius (GMed), Biceps Femoris (BF), Semitendinosus (ST), Gastrocnemius (medial) (GasMed), and Gastrocnemius (lateral) (GasLat), since the skin was prepared after shaving, alcohol cleaning, and drying. All sensors were installed according to the requirements outlined in SEMIAN guidelines [36]. Root Mean Square (RMS) data, which were acquired using a 30 ms window, were computed from the EMG signal after it had been transferred to EMGWorks Analysis™, Delsys, MA, USA. The EMG data for every muscle was normalized with the highest value found among the five USC tasks [37,38].

### 3.2. Kinematic Data

Sixty-four 19 mm single and cluster reflective markers were positioned to anatomic landmarks and segments for kinematic measurements (Table 1 and Figure 2).

Twelve Oqus 300 motion capture cameras recorded three-dimensional data at 500 frames per second through the Qualysis Track Manager™ (QTM, Goteborg, Sweden). The data was then exported to the Visual 3D v6 software for processing (C-motion™, Germantown, MD, USA) program. Winter [39] and Roewer et al. [40] showed that differences in movement artifact is reduced with a filter of 10 Hz for all the leg segments, so a 10 Hz Butterworth bi-directional (4th order) filter was applied to the kinematic data. For the analysis in the present paper, lower body data (hip and knee) were used in the sagittal and frontal planes at the beginning of contact and just before maximum knee flexion.

### 3.3. Kinetics

A Kistler force plate (Kistler Instruments Ltd.^®^, Model 569B, Winterthur, Switzerland) was used for ground reaction forces measurement, and the data were sampled at 1000 Hz. The 20 Hz Butterworth bi-directional (4th order) filter was applied to the movement data. All data was synchronized in QTM. Comparable to numerous studies, this filter level falls within the typical range of 12 Hz and 50 Hz [27,41].

### 3.4. Statistical Analysis

Data presented descriptively using means and standard deviations. The data was investigated using diagnostic statistics such at boxplots for outliers and Shapiro-wilk tests for normality. An independent samples *t*-test was employed to explore differences among group baseline characteristics. An ANOVA with three factors was performed in SPSS^®^ for the EMG data of every muscle: (1) a between-subjects factor: group; (2) a within-subjects factor: time before and after the landing; and (3) a within-subjects factor: the landing phase split between 30 ms before landing and 50 ms afterwards. If any main effects or interactions were significant, pairwise comparisons with Bonferroni corrections were carried out to identify differences. For variables such as maximum knee abduction, knee excursion, vGRF, normalized vGRF, and RFD, Two-Way mixed ANOVAs were employed by considering group as the between-subjects factor and time as the within-subjects factor. Analysis results were significant when the α-level was <5% (two-tailed). The authors mentioned the effect sizes (partial eta squared, ηp^2^) and pairwise test comparisons with 95% confidence intervals when appropriate.

## 4. Results

Between September 2016 and April 2017, a total of 51 people were recruited, and after the loss of 13 participants along the way (Figure 3), the final 38 participants (INT = 18, CON = 20) completed the study (Table 2). Most measured variables did not differ significantly when we compared the groups (Table 2); however, there was a significant difference in experience between the groups (t37, −2.2, *p* = 0.034, confidence interval (CI) = −3.72 to −0.155). There were no reported adverse events or injuries, and the overall compliance to the NMT program was 66.9%.

### 4.1. EMG

GasMed: There was no significant difference between the groups at 30 ms before IC (F (1,36) = 0.127, *p* = 0.723, ηp^2^ = 0.004). After 50 ms of IC, the amount of muscle activation increased in both groups (CON = 19.9, SD = 11.7 to 20.2, SD 10.4; INT = 25, SD 11.1 to 28.8, SD 10.8) and although non-significant it was higher in the INT (F (1,36) = 1.037, *p* = 0.315, ηp^2^ = 0.028) (Table 3).

GasLat: There was no significant difference 30 ms prior to landing (F (1,36) = 1.159, *p* = 0.298, ηp^2^ = 0.031). After 50 ms from the initial contact, there was a noticeable difference between the groups. Both groups showed more muscle activity (CON = 18.7% (SD 10.1) to 20.8% (SD 10.5); INT = 24.1% (SD 12.4) to 28.3% (SD 8.1), with a greater increase of 4.2% in the INT than the CON, rising by just 2.1% (F (1,36) = 1.844, *p* = 0.183, ηp^2^ = 0.490).

ST: There was no difference, whether interaction or main effects, in these muscles at 30 ms just before landing (F (1,36) = 1.149, *p* = 0.291, ηp^2^ = 0.031). There was a significant difference at 50 ms after landing (F (1,36) = 5.116, *p* = 0.030, ηp^2^ = 0.031) with both groups decreasing the muscle activation. The INT had a smaller reduction in EMG (CON = 6.2, INT = 3.3).

BF: Results from a mixed-design ANOVA demonstrated that there was no significant difference between the two groups at either point in time (F1,36) = 0.065, p = 0.800, ηp^2^ = 0.002, F (1,36) = 0.333, p = 0.567, ηp^2^ = 0.008 respectively). Both groups experienced approximately the same increase in muscle activation for both time points: CON from 30 ms before IC—from 29.1% (10.2) to 30.5% (17.0); INT pre-test from 32.3% (11.1) to 33.05% (11.3); CON from IC to 50 ms after—from 35.6% (11.1) at pre-test to 33.4% (9.9); INT from IC to 50ms after = 35.5% (12.2) at pre-test to 35.6 (9.6) at post-test.

GMed: There was a significant interaction effect for 30 ms before landing (F (1,36) = 6.759, p = 0.013, ηp^2^ = 0.290) and 50 ms after landing (F (1,36) = 4.810, *p* = 0.035, ηp^2^ = 0.118). Between the CON and INT, there was less reduction in muscle activation 30 ms before landing in the INT compared to the CON (2.6% vs. 6%), which was statistically significant (*p* = 0.013, F (1,36) = 6.759, ηp^2^ = 0.158). Following the landing, there was a drop of 4.7% in the CON but a similar or slight rise (0.7%) for the INT (F (1,36) = 4.810, *p* = 0.035, ηp^2^ = 0.118).

GMax: There was no significant difference at 30 ms before landing for this muscle, as both groups showed decreases in GMax, with the INT having a smaller reduction (~4%), less than the CON, which lowered it by nearly 5% (F1,36) = 3.577, *p* = 0.067, ηp^2^ = 0.090). There was no significant difference in post-landing electromyography at IC. (*p* = 0.542). Both groups saw an increase in muscle activation, with the INT increasing more, from 43.1% to 45.3% (SD 10.0 to 18.9), compared to the CON’s increase from 19.9% to 20.5% (SD 11.7 to 10.1) (F1,36) = 0.378, *p* = 0.542, ηp^2^ = 0.010).

### 4.2. EMG Results, Kinematics Results, Kinetics Results

No differences found in the sagittal or frontal planes of hip and knee measurements were found at initial or maximum knee flexion, when comparing groups or looking within groups, including maximum knee abduction (Table 4 and Table 5). Specifically, no significant differences were found in hip flexion (F (1,36) = 3.713, *p* = 0.062, ηp^2^ = 0.093;) or knee flexion (F (1,36) = 0.085, *p* = 0.772, ηp^2^ = 0.002) at IC. In addition, at IC in the frontal plane, the hip (Lateral Flexion—F (1,36) = 0.533, p = 0.47, ηp^2^ = 0.015 F (1,36) = 2.685, *p* = 0.110, ηp^2^ = 0.099) and the knee abduction (F (1,36) = 1.466, *p* = 0.234, ηp^2^ = 0.039) did not show any significant differences.

No significant differences were found for hip and knee flexion values at MKF (hip: F (1,36) = 3.401, *p* = 0.073, ηp^2^ = 0.086; knee: F (1,36) = 0.172, *p* = 0.680, ηp^2^ = 0.005). Similarly, the abduction/adduction movement at the hip (F (1,36) = 0.835, *p* = 0.367, ηp^2^ = 0.023) and the knee (F (1,36) = 4.308, *p* = 0.055, ηp^2^ = 0.107) did not significantly differ.

However, regardless of the lack of statistically significant differences, there were some trends worth noting in maximum knee abduction and knee excursion after the 8 weeks (Figure 4). There was a significant within-subjects main effect for maximum knee abduction (F (1,36) = 7.721, *p* = 0.009, ηp^2^ = 0.18), as well as a substantial time-by-group interaction (F (1,36) = 8.096, *p* = 0.007, ηp^2^= 0.18). The INT group experienced a decrease in maximum knee abduction after eight weeks of Neuromuscular Training (NMT) (Week 8: INT Mean = 8.47, SD = 2.8) whereas the control group experienced an increase.

### 4.3. Kinetics

Peak vGRF and normalized vGRF showed no significant difference between the groups (F (1,36) = 0.042, *p* = 0.838, ηp^2^ = 0.001; F (1,36) = 0.408, *p* = 0.527, ηp^2^ = 0.011) (Table 6). The vGRF peak increased by more than 40 N in the CON group, and it decreased by about 55 N in the INT group.

*RFD:* The mean RFD was similar between the groups at pre-test (CON = 19.5 (9.9) vs. INT = 18.9 (9.1) BW/s). However, there was a significant time by group interaction; the mean RFD for the CON increased (by 2.22 BW’s/s), whereas the mean for the INT decreased (by 6.08 BW’s/s). At week 8: CON = 21.75 BWs/s, SD = 9.4; INT = 12.83 BWs/S, SD = 5.7) (within subjects main effect: F (1,36) = 2.476, *p* = 0.124, ηp^2^ = 0.064; between groups effect: F (1,36) = 3.593, *p* = 0.066, ηp^2^ = 0.091; time by group interaction: (F (1,36) = 11.519, *p* = 0.002, ηp^2^ = 0.242) (Figure 5).

## 5. Discussion

The study sought to explore how a NMTP could improve the movement skills of female hockey players. According to the hypothesis, neuromuscular training could potentially increase the activation profile in lower limb muscles, which could lead into more controlled movements and subsequently the players to be subjected in lower risk of non-contact injuries.

### 5.1. EMG

After the 8 weeks period, the normalized gluteus maximus (GMax) EMG was largely unchanged in both groups despite the inclusion of squats, lunges, and glute bridge exercises in the intervention, all of which have been reported to elicit GMax activation [42]. The gluteus medius (GMed) activation decreased in the CON, whereas the INT maintained the muscle activation. Although the intervention included hip abduction movements shown to produce gluteus medius muscle activation by DiStefano et al. [16] and Boren et al. [43], future interventions may benefit from more specific exercises to increase activation and therefore reduce hip adduction during landing in high intensity sports activities. These findings differ from Weir et al. [17] who reported an increase of 10% total gluteal muscle activation. This might be due to the population differences between the two studies (i.e., elite vs. recreational hockey players), the additional strength and conditioning sessions the elite players usually complete, the length of the intervention or a combination of these factors. In addition, the program followed by Weir et al. might have featured a greater proportion of glute specific exercises and measured total gluteal activity whereas this study included workouts for the whole lower limb.

The gluteal muscles are responsible for key movements such as hip extension, abduction and external rotation and hence for opposing adduction and internal rotation which are usually involved to common injury mechanisms [35,44,45]. The NMTP in this study included single-leg squats, arabesques, lunges, and plyometrics to assist this group of muscles. However, no significant increase in GMed activation was found after the intervention, which could suggest that more frontal plane specific exercises should be considered in future NMTP or the overall training volume and intensity of the programme needs further consideration.

The changes in hamstring muscle activation were not statistically significant when comparing baseline and outcome timepoints, even with the inclusion of Nordic hamstring curls and other eccentric contractions during the latter stages of the swing phase of sprinting and arabesque, all of which are high hamstring muscle activation exercises [46] and reduce injury rates [47]. A study by Zebis et al. [48] revealed that athletes involved in team sports experience increased ST activation after NMTP. Zebis et al. found there was no difference between INT and CON at pre-test in Semitendinosus activation however, there was a significant difference at post-test as the activation in the CON decreased whereas the INT increased the activation of this muscle. These results are different from our study possibly because the program duration (12 weeks vs 8 weeks), and maybe due to the lack of progression in this study. In addition, Zebis et al. [49] found that prompting the ST system before IC with a lead time of 30 ms is essential for using safe movement techniques to protect the ACL., Further investigation may be warranted to find the optimum training volume, training progression, for increased muscle hamstring activation.

There was an increase in the GasMed and GasLat activity between the groups from IC to 50 ms post-landing at week 8, with a greater increase in the INT (a significant difference at week 8). As the gastrocnemius helps control, along with other lower extremity muscles, both the knee and ankle [50,51], this increase may be important. This change may be due to the muscle activation exercises at the start of the NMTP, agility and plyometric exercises directly, as well as the calf raises, lunges, running, and sport-specific movement along with the advised change in technique. These exercises can improve both activation and gastrocnemius reactivity [52], as there is considerable loading and unloading, which may not have occurred in the control group. The increase in gastrocnemius activity, especially GasMed [53] with associated knee and ankle control, could help reduce the risk of injuries, as both joints are frequently injured in hockey [2,4].

Brunner et al. [54] and Zebis et al. [48] propose NMTP with a frequency of at least twice a week for six weeks. The researchers suggest that, for this type of training to be effective an increase in the dosage to four times per week is required. Furthermore, future studies may benefit from monitoring intensity, effort and commitment to the intervention similar to Weir et al. [17]. in this study may have been influenced by the degree of attention the participants gave and the effort they put into their workouts. Using EMG alone is not the only way to find out if muscles are changing for the better. Also, the intervention could be developed to have a hamstring focus such as more Nordic Hamstring Exercises, which have been shown to strengthen and stimulate hypertrophy [55].

### 5.2. Kinematics

There were no significant differences in the two groups’ sagittal and frontal plane movements of the hip. Decreased hip abductor strength such as the gluteal muscles, can cause excessive hip movement. [56,57]. It has been reported that the hip mobility may be altered after NMT, specifically a 3-month strength-based NMTP can decrease the hip adduction and knee abduction during the single-leg triple hop (Baldon et al. [58]. Evidence indicates that hip adduction is related to knee abduction during landing [27,59]. The length of the intervention and nature of the task may limit the change in frontal plane motion.

The INT and CON had comparable movements in the sagittal plane at MKF and IC, respectively. This outcome may come from the participants receiving additional feedback from outside sources as part of the intervention. Using feedback through communication appears to be an optimal way to develop and use skills in playing sports [32]. Moreover, when augmented feedback is used in training, there has been a reduction in knee valgus in the frontal plane to reduce more compared to no feedback (37.23% vs. 26.7%) [60] and this is important since knee abduction significantly raises the risk for ACL injury [26,60,61]. Recent studies indicate that nearly all ACL injuries in football players were caused by movements in the knee with valgus loading. During tasks with no contact, valgus loading occurred at the knee for 67% of kicks and 50% of jumps [62]. Hopper et al. [63] indicate that after neuromuscular training, female netball players increased knee flexion at both IC and MKF, unlike the control group, who had decreased knee flexion. This may be due to the focus (all lower body exercises) of the NMTP in their study, additionally, there was feedback on the players technique.

### 5.3. Kinetics

A trend of reduced vGRF following NMT in the INT, with a slight increase in the CON during the USC was observed in this study which could imply that CON are at greater injury risk [63,64]. In addition, a strong relationship exists between vGRF and the knee abduction motion since it has been reported to be 20% higher in female athletes with ACL injuries [27]. The absolute vGRF for the CON in this study was similar to that of the injured cohort in Hewett et al.’s [27] study, however it fell far below the ACL’s tension limit [65]. In addition, a decrease in vGRF (with greater knee flexion) can reduce the knee flexion–extension moment and therefore anterior tibial shear [66]. This may be due to the exercises and partly because of the augmented feedback the athletes received throughout the study since it has need reported that the use of feedback can enhance motor learning and reduce the risk of injury [32] Therefore, the trends in this study suggest that the holistic NMTP which was implemented for an optimal period can potentially enhance motor learning, optimize loading profiles in activities which involve landing and change of direction and could potentially reduce the risk of injury in athletes.

The RFD was significantly lower in the INT vs. CON (21.8 vs. 12.8 BWs/s at post-test). This is in line with another study using a NMTP [67] and might be an indication that active structures (i.e., muscles) can potentially absorb more energy than the passive structures (i.e., ligaments and joints). Therefore, switching the traditional warm-ups (usually involving sets of 2 min pulse raisers, stretching, low intensity hockey skills, and some specific team-play maneuvers) to a NMTP could possibly lead to valuable outcomes. It should be underlined that the landing strategy is very important in reducing the RDF and a heel-first landing strategy can significantly increase (~33%) the loading and by [68]. In this study, particular attention was given to providing feedback and coaching to the athletes to improve their landing technique as suggested by Benjaminse et al. [32].

### 5.4. Limitations and Future Studies

Since the study was not a randomized controlled trial (RCT), there is a possibility of bias and a type 2 error. Future studies could be conducted as an RCT to enhance internal validity. Although a power calculation was conducted in this study, a larger sample size could potentially improve the power of the study to detect existing differences. The attendance was monitored in this study the intensity was not measured (e.g., measuring heart rate or rate of perceived exertion). This measure could be included in future studies to monitor each session of the intervention, since it is well known that the training intensity is an important factor in training programs. The dosage effect by increasing the amount of time and effort a person devotes to the program might help their EMG, kinematics, and kinetics improve. In addition, the participants in this study were recreational players compared to trained elite athletes that are participating in other studies which makes comparing the datasets difficult.

Future studies could include a more targeted approach on gluteal and hamstring muscles, as there were limited activation changes. In addition, future studies may benefit from progressive overload throughout the intervention. Furthermore, this study focused on recreational hockey players and therefore, future studies could include elite hockey players or further investigation of the intervention with other team sport players.

## 6. Summary

Following 8 weeks of NMTP, in this study trends of higher muscle activation were observed especially for the gluteal and gastrocnemius muscles. Moreover, the findings include a reduction of the knee abduction during landing and the overall knee excursion during the USC. Finally, a more marked reduction of the vertical ground reaction force and rate of force development in the NMTP group as compared to the control group was also observed. Altogether, these findings might suggest improved motor control and hence possibly lower risk of sports related injury after the implementation of a NMTP in recreational hockey players. For this reason, further research in the design and implementation of evidence based NMTP in team sports is warranted.

## Figures and Tables

**Figure 1 bioengineering-12-01101-f001:**
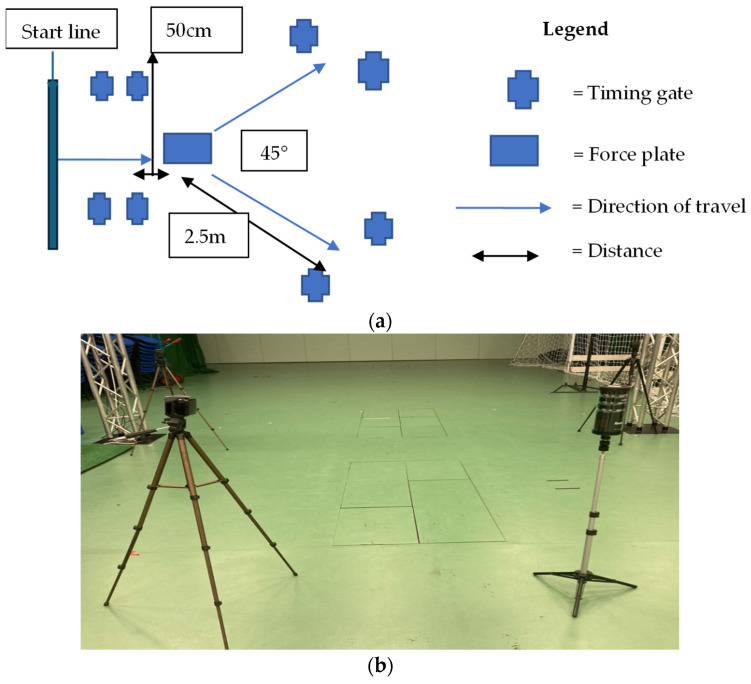
The set-up for the unanticipated side-cut task (**a**). The set-up of the equipment in the laboratory (**b**).

**Figure 2 bioengineering-12-01101-f002:**
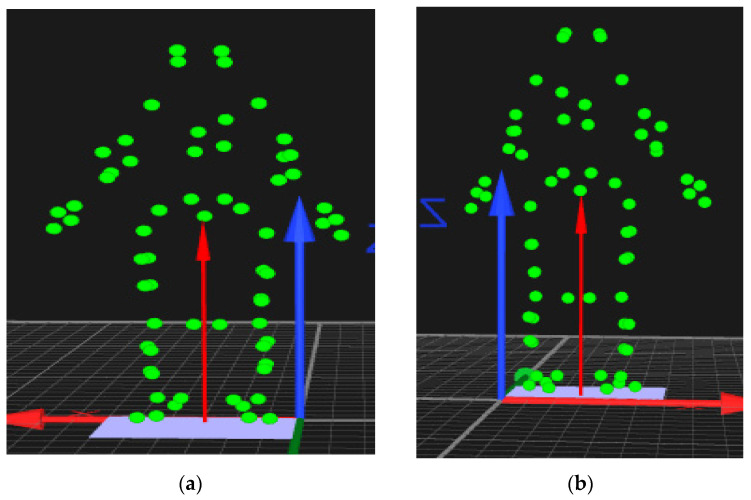
**Green dots represent the** reflective marker placement—(**a**) anterior view and (**b**) posterior view (depicted by Qualisys). Arrows show the global laboratory Axis reference system and the resultant ground reaction force.

**Figure 3 bioengineering-12-01101-f003:**
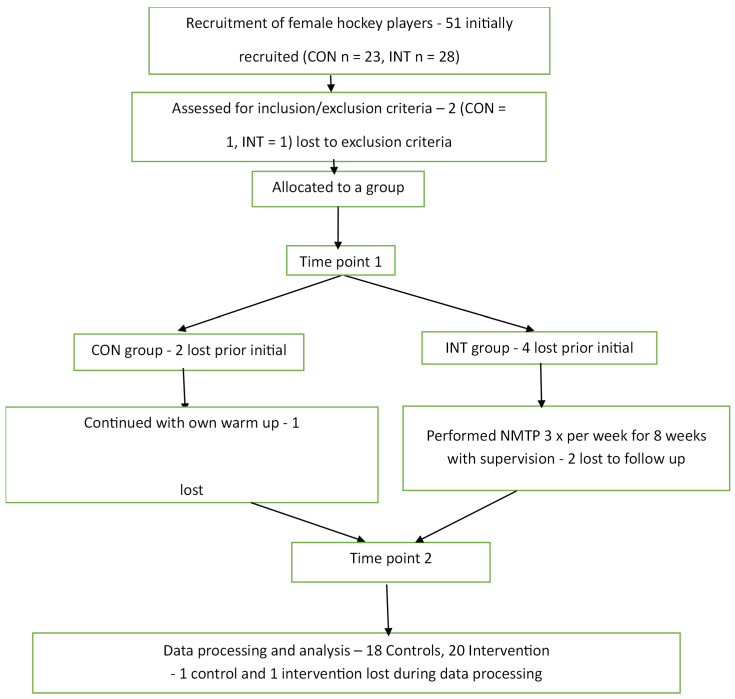
Flow diagram of participants through the study (CON—control; INT—intervention; NMTP—neuromuscular training program).

**Figure 4 bioengineering-12-01101-f004:**
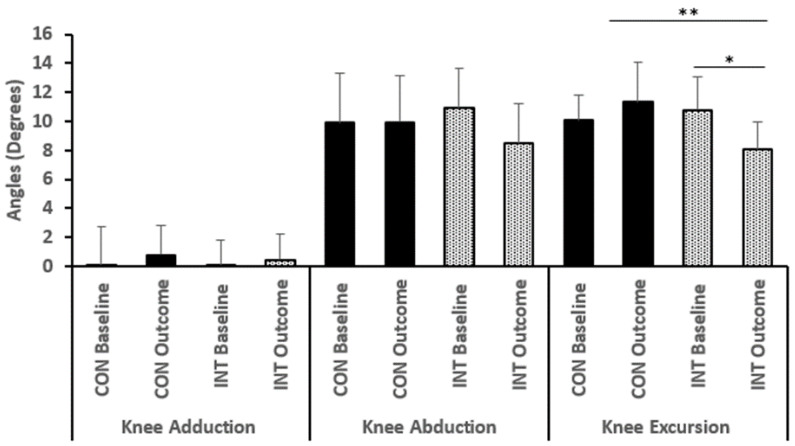
Maximum knee adduction, abduction, and excursion angles during an unanticipated side-cut (means ± SD). ** Significant interaction effect, * Significant within subjects main effect. INT Intervention, CON: control.

**Figure 5 bioengineering-12-01101-f005:**
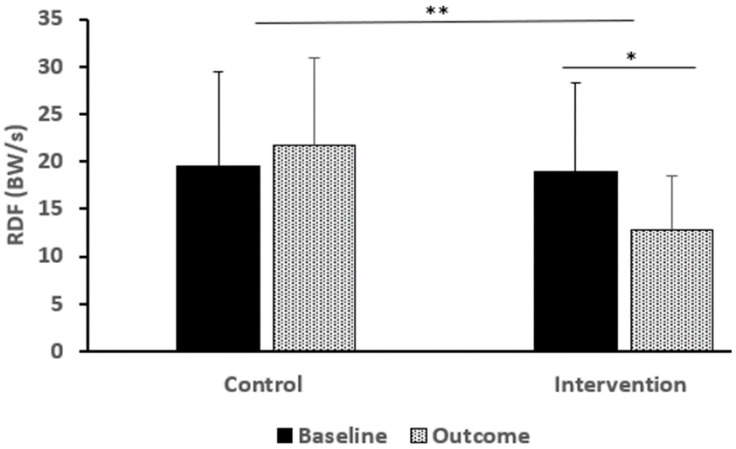
Rate of Force Development (RDF) during an unanticipated side-cut (body weight per second, BW/s) (means ± SD). ** Significant interaction effect, * Significant within subjects main effect. INT Intervention, CON: control.

**Table 1 bioengineering-12-01101-t001:** Reflective marker placement.

Anatomical Site	Marker Placement (No. of Markers)
Head	Left and Right Anterior and Posterior Cranium (4)
Trunk (Treated as a Single Segment)	Acromion Process (2) Posterior Superior Iliac Spine—PSIS (2) Upper Thoracic Cluster (4)
Arm and Finger (Bilateral)	Acromion Process (1) Lateral and Medial Elbow Epicondyle (2) Lateral and Medial Aspects of the Styloid Process (2) Third Metacarpal (Distal End) (1)
Pelvis (a CODA Pelvis)	Anterior Superior Iliac Spine—ASIS (2) Posterior Superior Iliac Spine—PSIS (2) Sacrum
Thigh (Bilateral)	Greater Trochanter (1) Lateral and Medial Femoral Condyle (2) Thigh Cluster (4)
Shank (Bilateral)	Lateral and Medial Femoral Condyle (2) Lateral and Medial Malleolus (2) Shank Cluster (4)
Foot (Bilateral)	Lateral and Medial Malleolus (2) Heel (1) 1st and 5th Metatarsal (2)

**Table 2 bioengineering-12-01101-t002:** Participant characteristics (Mean ± SD).

Variables	Groups		
	CON	INT	*p*-Value
Age (yrs)	20.04 (1.6)	19.9 (1.1)	0.97
Height (cm)	165.2 (4.7)	167.6 (5.4)	0.40
Body mass (kg) (baseline)	62.9 (7.8)	66.0 (6.3)	0.68
Body mass (kg) (8 weeks)	62.7 (7.8)	66.4 (6.4)	0.74
No. of games per week (n)	1.5 (0.6)	1.8 (0.4)	0.09
No. of training sessions (hockey) (n)	1.21 (0.4)	1.0 (0.3)	0.92
No. of training sessions per week (not hockey) (n)	1.0 (0.6)	1.1 (0.8)	0.43
Playing experience (years)	9.3 (3.1)	11.2 (2.4)	0.03
No. of injuries in intervention period (n)	0.2 (0.4)	0.1 (0.3)	0.32

**Table 3 bioengineering-12-01101-t003:** Normalized EMG during the unanticipated side-cut.

Muscle	Time Point	Group	Normalized EMG (%) Mean (SD)		*p*-Value
			Baseline	Week 8	Interaction
GasMed	30 ms prior to IC	CON INT	25.3 (14.9) 18.3 (11.6)	28.4 (11.9) 23.1 (16.7)	0.723
	50 ms after IC	CON INT	19.9 (11.7) 25.1 (11.1)	20.2 (10.4) 28.8 (10.8)	0.315
GasLat	30 ms before IC	CON INT	27.1 (15.2) 17.9 (9.9)	25.9 (9.9) 24.0 (15.0)	0.298
	50 ms after IC	CON INT	18.7 (10.1) 24.1 (12.4)	20.8 (10.5) 28.3 (8.1)	0.183
ST	30 ms before IC	CON INT	35.9 (11.5) 35.7 (15.1)	31.1 (9.9) 34.2 (18.6)	0.291
	IC to 50 ms	CON INT	37.2 (11.6) 37.8 (11.5)	31.1 (8.2) 34.5 (9.8)	0.030
BF	30 ms before IC	CON INT	36.3 (13.7) 33.7 (15.2)	37.1 (15.2) 34.4 (11.4)	0.800
	IC to 50 ms	CON INT	35.6 (11.1) 35.5 (12.2)	33.4 (9.9) 35.6 (9.6)	0.567
GMed	30 ms before IC	CON INT	27.9 (11.7) 24.7 (7.1)	21.9 (6.1) 22.1 (8.4)	**0.013**
	IC to 50 ms	CON INT	29.2 (10.8) 23.2 (8.2)	24.5 (6.1) 23.9 (4.7)	**0.035**
GMax	30 ms before IC	CON INT	29.2 (12.6) 28.9 (9.8)	24.4 (9.6) 24.9 (12.0)	0.067
	IC to 50 ms	CON INT	19.9 (11.7) 27.2 (11.9)	20.5 (10.1) 29.0 (10.6)	0.542

*(GasMed = Gastrocnemius (medial), GasLat = Gastrocnemius (Lateral), ST = Semitendinousus, BF = Biceps Femoris, IC = Initial Contact, millisseconds, CON = control group, INT = intervention group).*

**Table 4 bioengineering-12-01101-t004:** Lower extremity sagittal and frontal plane kinematics at initial contact (IC).

Joint	Time Point	Variable	Group	Value		
				Before	Week 8	Interaction
Hip	IC	Flexion	CON INT	38.86 (8.6) 41.99 (9.8)	37.85 (8.7) 35.77 (8.7)	0.853
	IC	Lateral flexion	CON INT	12.74 (5.4) 10.87 (6.4)	10.87 (6.4) 9.72 (5.4)	0.291
Knee	IC	Flexion	CON INT	20.70 (6.7) 19.28 (10.1)	18.64 (8.4) 17.96 (8.2)	0.193
Knee	IC	Abduction	CON INT	0.27 (4.5) 1.01 (4.3)	0.77 (4.9) 1.9 (3.1)	0.732

**Table 5 bioengineering-12-01101-t005:** Lower extremity sagittal and frontal plane kinematics at maximum knee flexion (MKF).

Joint	Time Point	Variable	Group	Value		
				Before	Week 8	Interaction
Hip	MKF	Flexion	CON INT	45.7 (11.7) 45.2 (11.1)	46.1 (11.0) 38.8 (8.0)	0.073
	MKF	Lateral flexion	CON INT	13.02 (7.3) 11.4 (6.6)	12.4 (7.2) 10.4 (6.2)	0.370
Knee	MKF	Flexion	CON INT	61.8 (7.2) 62.0 (11.5)	63.6 (6.9) 63.6 (6.8)	0.680
	MKF	Abduction	CON INT	−1.09 (6.7) −3.81 (7.6)	−3.65(7.1) −5.68 (6.6)	0.055

**Table 6 bioengineering-12-01101-t006:** Peak vertical ground reaction force (vGRF) and normalized (to body weight) vertical ground reaction force (Norm vGRF).

Variable	Group	Value Mean (SD)		*p*-Value		
		Before	Week 8	Interaction	Within	Between
Peak vGRF	CON	1185.2 (212.6)	1226.3 (211.2)	0.838	0.170	0.910
(N)	INT	1226.7 (208.6)	1171.5 (180.5)			
Norm vGRF	CON	1.97 (0.3)	1.98 (0.3)	0.527	0.360	0.100
(N/BW)	INT	1.90 (0.3)	1.82 (0.2)			

## Data Availability

The data presented in this study are available on request from the corresponding author. The data are not publicly available due to ethical considerations.

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
