# Peer review of "The Effects of Neuromuscular Training on Electromyography, Lower Extremity Kinematics, and Ground Reaction Force During an Unanticipated Side-Cut on Recreational Female Hockey Players"

_bioengineering, 2025, doi:10.3390/bioengineering12101101_

Round 1

Reviewer 1 Report

Comments and Suggestions for Authors

1. The introduction lacks a clear statement of the contributions and the overall structure of the paper.
2. The summary section is overly brief and should be expanded to better highlight the key findings and implications of the work.

Author Response

1. The introduction lacks a clear statement of the contributions and the overall structure of the paper.

I have amended the manuscript - included a statement on how this paper contributes to the literature. to account for this comment. 

2. The summary section is overly brief and should be expanded to better highlight the key findings and implications of the work.

The summary section has been expanded on to clearly show the results of this study.

Reviewer 2 Report

Comments and Suggestions for Authors

The article is very valuable in terms of being specific to a specific sport and sports technique.
Introduction: New studies 2020 to 2025 not used
Material and methods : You can use more figures, diagrams, and tables, especially figures.
Your most important variable is the intervention and type of neuromuscular training, which should be explained more precisely and in detail.
In addition, the amount, intensity, and dosage of the exercise and how to increase it in 8 weeks should be explained.(FITT?)
Of course, the type of neuromuscular training requires a period of 3 months, unlike strength training, which is longer than 8 weeks.
Given that neuromuscular training has the greatest impact on muscle timing, why didn't you measure this variable?
Did the anthropometric status, especially these variables in the lower limbs and pelvis, have an effect on the measured variables? If it had an effect, why was it not measured and compared between the two groups?
DISCUSSION: In the discussion to compare the results of your work with other researchers, you did not compare your work with others in terms of the type of intervention and exercises.
Summary : You cannot conclude from the results that these changes seen can prevent damage or reduce the chance of it occurring.

Author Response

The article is very valuable in terms of being specific to a specific sport and sports technique.  Introduction: New studies 2020 to 2025 not used

I have included more of the literature that falls within that date in the introduction. 

Material and methods : You can use more figures, diagrams, and tables, especially figures.

I have included more figures to clarify procedure. 

Your most important variable is the intervention and type of neuromuscular training, which should be explained more precisely and in detail.

I have include supplementary information to detail intervention. 

In addition, the amount, intensity, and dosage of the exercise and how to increase it in 8 weeks should be explained.(FITT?)

Within the intervention there is information on how to increase load.

Given that neuromuscular training has the greatest impact on muscle timing, why didn't you measure this variable?

This is a good point. I did initially investigate this however the factors/variables (threshold etc) have a large impact on the results, therefore I didn't pursue this aspect of EMG.

Did the anthropometric status, especially these variables in the lower limbs and pelvis, have an effect on the measured variables? If it had an effect, why was it not measured and compared between the two groups?

This comment is fair. However, there were no differences (apart from playing experience) between the groups. 

In the discussion to compare the results of your work with other researchers, you did not compare your work with others in terms of the type of intervention and exercises.

I have included some information on the intervention and differences to account for this comment.
Summary : You cannot conclude from the results that these changes seen can prevent damage or reduce the chance of it occurring.

I have altered the Summary to take on board this comment.

Reviewer 3 Report

Comments and Suggestions for Authors

The authors report an investigation into the effects of neuromuscular training during an unpredictable side-cut in 38 recreational female hockey players. The authors analyze the effects on electromyography response, lower extremity kinematics, and ground reaction force.

The article is well-written, and what's noteworthy is that it compares the results with those reported by several authors. It finds similarities and reports some differences, justifying the reasons for its results.

The work would be excellent if it included some graphs of the myoelectric signals comparing the most relevant results of the two groups considered (intervention group (INT) and control group (CON)).

The work would be excellent if some graphs of the myoelectric signals were included, comparing the results of the two groups considered (intervention group (INT) and control group (CON)).

Also, to highlight it, it would be advisable to include a photo of a user showing the equipment or performing some test.

Author Response

The authors report an investigation into the effects of neuromuscular training during an unpredictable side-cut in 38 recreational female hockey players. The authors analyze the effects on electromyography response, lower extremity kinematics, and ground reaction force.

Thank you for this comment. No action required.

The article is well-written, and what's noteworthy is that it compares the results with those reported by several authors. It finds similarities and reports some differences, justifying the reasons for its results.

Thank you for this comment. No action required.

The work would be excellent if it included some graphs of the myoelectric signals comparing the most relevant results of the two groups considered (intervention group (INT) and control group (CON)). The work would be excellent if some graphs of the myoelectric signals were included, comparing the results of the two groups considered (intervention group (INT) and control group (CON)).

I did consider this however as the EMG signal is normalised. It was considered that the was little value in this figure. 

Also, to highlight it, it would be advisable to include a photo of a user showing the equipment or performing some test.

I have included a photo of the task in this investigation. 

Reviewer 4 Report

Comments and Suggestions for Authors

Thank you for submitting to bioengineering. The topic is appropriate and the content of the experiment is positive. However, the author should write in detail and improve it to introduce the research more kindly to the readers.

Introduction
Overall, the logical flow is weak. The introduction needs a new structure to clearly show the purpose of the study. The author's introduction states that the purpose of NMT is injury prevention. Since the gluteus medius and knee valgus are among the many factors causing injuries, I hope you will approach it with comprehensive information. In line 41, the content of 'Mixing plyometrics, training with weights' suddenly appears. The author needs content to connect the previous sentence and this part.
Therefore, I recommend the following structure of the introduction according to the title:
Injuries and causes of hockey players - Introducing training methods to prevent injuries - Effects and results of the introduced training methods - Shortcomings of previous studies (There is a lack of research on female hockey players, Unanticipated Side-Cut research is needed, etc.) - Therefore… .

Research method: The research method needs to be written in detail overall.

Need to write in detail about how players are recruited. For example, internet, bulletin boards, during or off season.
Intervention: Please write in detail about training. Time, frequency, intensity, method, posture, rest time, rest period, movement guidance and monitoring, etc. It would be great if the author could add movement photos.
Merge sections 2.2 and 2.3.
Add photos related to Unanticipated Side-Cut and photos related to the measurement setup to show more details.

Results:
Table 2: Include Effect size and t value to help readers understand. Also record Con (n=?), INT (n=?).
Figure 4: Can the knee excursion be modified to another anatomical movement term? Because it is inconsistent with knee adduction, knee abduction.
Match the decimals. (Table 3: Suddenly 0.1, 0.52 in GMed is not good).
And please make the table editing a little easier to read. Ex) Table 3: Make 30ms prior to IC appear in one line. And some are prior and some are before. Please write more carefully.
Table 4: Match the position of the text. Ex) 0.061 and 0.85 are in different positions. (The same goes for other tables).
Figure: You can increase the resolution and reduce the size of the figure a little. Make the text of the figure similar to the text size of the main text.

Discussion
What is the author's training intention? "Controlling the valgus of the knee can prevent injury"?
The discussion should sufficiently compare the biomechanical analysis of other studies with the analysis of this study. And since the EMG test was conducted, there should be a discussion about which muscle activation and stimulation provide greater benefits and which muscle control plays a major role.

The citation and referencing style does not follow the MDPI format. Authors are requested to adhere to the MDPI guidelines. And there are many old documents.

Author Response

Thank you for submitting to bioengineering. The topic is appropriate and the content of the experiment is positive. However, the author should write in detail and improve it to introduce the research more kindly to the readers.

I have re-structured the introduction to introduce the subject more kindly. 

Overall, the logical flow is weak. The introduction needs a new structure to clearly show the purpose of the study. The author's introduction states that the purpose of NMT is injury prevention. Since the gluteus medius and knee valgus are among the many factors causing injuries, I hope you will approach it with comprehensive information.

Thank you for the feedback. I have re-structured to account for this comment.  

In line 41, the content of 'Mixing plyometrics, training with weights' suddenly appears. The author needs content to connect the previous sentence and this part.

Thank you for this feedback. I have re-worded to make the sentence flow better. 

Therefore, I recommend the following structure of the introduction according to the title:
Injuries and causes of hockey players - Introducing training methods to prevent injuries - Effects and results of the introduced training methods - Shortcomings of previous studies (There is a lack of research on female hockey players, Unanticipated Side-Cut research is needed, etc.) - Therefore… .

Thank you for this structure. I have used it to help the readers. 

Research method: The research method needs to be written in detail overall.

Than you for this comment.  I have added more detail and a figure to increase information. 

Need to write in detail about how players are recruited. For example, internet, bulletin boards, during or off season.

Thank you for this comment. I have added information accordingly. 

Intervention: Please write in detail about training. Time, frequency, intensity, method, posture, rest time, rest period, movement guidance and monitoring, etc. It would be great if the author could add movement photos.

I have included supplementary information to provide details of the intervention. 

Merge sections 2.2 and 2.3.

These sections have now been merged. 

Add photos related to Unanticipated Side-Cut and photos related to the measurement setup to show more details.

A photo has now been included. 

Results:
Table 2: Include Effect size and t value to help readers understand. Also record Con (n=?), INT (n=?).

Numbers have been added. 

Figure 4: Can the knee excursion be modified to another anatomical movement term? Because it is inconsistent with knee adduction, knee abduction.

The terms have been made more consistent in the terminology. 

Match the decimals. (Table 3: Suddenly 0.1, 0.52 in GMed is not good).

All decimal places are now the same. 

And please make the table editing a little easier to read. Ex) Table 3: Make 30ms prior to IC appear in one line. And some are prior and some are before. Please write more carefully.

This table has been altered to be more clear. 

Table 4: Match the position of the text. Ex) 0.061 and 0.85 are in different positions. (The same goes for other tables).

The table has been altered to be more clear.

Figure: You can increase the resolution and reduce the size of the figure a little. Make the text of the figure similar to the text size of the main text.

This has been done to more clear. 

What is the author's training intention? "Controlling the valgus of the knee can prevent injury"?

There is now more information to account for this comment. 

The discussion should sufficiently compare the biomechanical analysis of other studies with the analysis of this study. And since the EMG test was conducted, there should be a discussion about which muscle activation and stimulation provide greater benefits and which muscle control plays a major role.

This section has been altered to account for this comment. 

The citation and referencing style does not follow the MDPI format. Authors are requested to adhere to the MDPI guidelines. And there are many old documents.

The referencing section has been updated.

Round 2

Reviewer 4 Report

Comments and Suggestions for Authors

I do not  have any comments.

Author Response

Strengths 1- 5. Thank you for your comments.

Areas for improvement.

  1. Randomization and Blinding: Comment accepted. Information on future directions included.
  2. Training Dose and Compliance. Comment accepted. A limitation included to recognised this and provide directions for further studies. 
  3. Limited Generalizability: Comment accepted. Additional information has been included in the discussion section to recognised this.
  4.  Muscle Activation Specificity. Comment accepted. Additional information has been included to recognise limited muscle activation of the muscles mentioned. 
  5. Writing and Clarity. Comment accepted. Manuscript has been re-read and clarity added including those sentences mentioned by the academic Editor.